# Causal Discovery Under a Confounder Blanket

**David S. Watson**[1]

**Ricardo Silva**[1]

[1]Department of Statistical Science, University College London, London, UK

## Abstract

Inferring causal relationships from observational data is rarely straightforward, but the problem is especially difficult in high dimensions. For these applications, causal discovery algorithms typically require parametric restrictions or extreme sparsity constraints. We relax these assumptions and focus on an important but more specialized problem, namely recovering the causal order among a subgraph of variables known to descend from some (possibly large) set of confounding covariates, i.e. a *confounder blanket*. This is useful in many settings, for example when studying a dynamic biomolecular subsystem with genetic data providing background information. Under a structural assumption called the *confounder blanket principle*, which we argue is essential for tractable causal discovery in high dimensions, our method accommodates graphs of low or high sparsity while maintaining polynomial time complexity. We present a structure learning algorithm that is provably sound and complete with respect to a so-called *lazy oracle*. We design inference procedures with finite sample error control for linear and nonlinear systems, and demonstrate our approach on a range of simulated and real-world datasets. An accompanying R package, `cbl`, is available from `CRAN`.

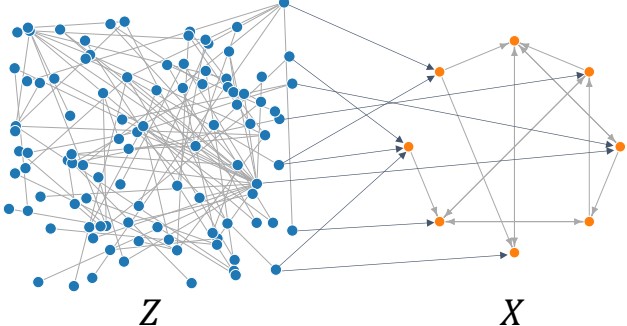

Figure 1: Visual depiction of our setup, which includes a large collection of background variables $Z$ (blue nodes) with arbitrary structure, followed by a relatively small set of foreground variables $X$ (orange nodes). The goal is to learn causal relationships among $X$ variables by exploiting signals from $Z$.

## 1 INTRODUCTION

Discovering causal relationships between variables is a vital first step in any effort to understand complex systems or design effective interventions. In principle, such relationships can be established through sufficient experimentation; in practice, we must often make do with observational data due to logistical or ethical constraints. Causal discovery algorithms have been in use for decades—see [Glymour

et al., 2019] for a recent review—but the task is notoriously difficult and error-prone, especially in high-dimensional settings. Moreover, many of these methods, for computational or statistical tractability, assume *scale-free sparsity*—i.e., that the number of adjacencies for each vertex in the true graph does not grow with the dimensionality of the problem.

In many cases, researchers are interested primarily in the causal relationships between just a subset of observed variables. Attempting to learn an entire directed acyclic graph (DAG) in such cases is inefficient and unstable, especially when error rate control is a concern and unmeasured confounders cannot be ruled out. Suppose, however, that we have access to a large tier of background factors $Z$ that may potentially deconfound our target system $X$. This stratification could be due to temporal ordering or physical laws. For example, we know that genotypes precede phenotypes, even though it may be impossible to completely characterize the relationship between the two, let alone links among the genotypes themselves. We argue that many practical problems of interest exhibit such a two-tier structure, with our

*Accepted for the 38th Conference on Uncertainty in Artificial Intelligence* (UAI 2022).

*foreground* variables $\boldsymbol{X}$ causally preceded by some large *background* set $\boldsymbol{Z}$, whose internal structure is not relevant or even well-defined.

We propose a novel structure learning algorithm designed for such setups. Our method leverages "pre-system" background covariates $\boldsymbol{Z}$ to establish causal relationships among foreground variables $\boldsymbol{X}$ without making any assumptions about the sparsity of connections between the two tiers. The trade-off is that it will not attempt to discover every possible structural signature that a typical causal discovery method can in theory resolve [Spirtes et al., 2000]. Instead, we limit ourselves to what can be derived from the background-foreground interaction. In particular, we posit that background variables can act as a *confounder blanket*, which, as a whole, either blocks unmeasured confounding or not. This amounts to a bet that we can avoid combinatorial search over subsets of $\boldsymbol{Z}$ and still get informative results.

Our main contributions are threefold. (1) We derive a sound and complete set of rules for inferring causal order in subgraphs with background variables, as well as identifiability conditions for causal discovery in such settings. Completeness is derived with respect to a so-called *lazy oracle*, which we argue is of greater practical relevance in many settings than the classical independence oracle, especially when we are concerned about statistical and computational feasibility. (2) We design an algorithm that implements these rules with finite sample error control, making a further assumption about how to test for statistical independencies based on regression models. The method is efficient and flexible, avoiding the combinatorial search associated with alternative methods and accommodating both linear and nonlinear systems. (3) We test our approach against a range of alternatives on simulated and real-world data, confirming that the method recovers ancestral relationships in the target subgraph with high power and bounded error.

## 2  BACKGROUND AND NOTATION

We assume that causal relationships can be encoded as a DAG $\mathcal{G}$. Each vertex in $\mathcal{G}$ represents a random variable in a distribution with density/mass function $p(\cdot)$. We make use of the following common terminology in causal discovery: *parent*, *child*, *ancestor*, *descendant*, *mediator*, *collider*, *(active/backdoor) path*, *d-separation*, and *Markov equivalence class*. We omit formal definitions due to space constraints. For details, see [Spirtes et al., 2000, Pearl, 2009a].

We use $\boldsymbol{X} \perp\!\!\!\perp_{\mathcal{G}} \boldsymbol{Y} \mid \boldsymbol{Z}$ to denote that set $\boldsymbol{X}$ is $d$-separated from set $\boldsymbol{Y}$ given set $\boldsymbol{Z}$ in $\mathcal{G}$. The notation is deliberately similar to that of conditional independence in probability theory [Dawid, 1979]. We stipulate that the joint distribution of the data is Markov with respect to $\mathcal{G}$, i.e. $\boldsymbol{X} \perp\!\!\!\perp_{\mathcal{G}} \boldsymbol{Y} \mid \boldsymbol{Z} \Rightarrow \boldsymbol{X} \perp\!\!\!\perp \boldsymbol{Y} \mid \boldsymbol{Z}$. When the distribution is *faithful* to the graph, the converse holds as well, upgrading the relationship to a

biconditional: $\boldsymbol{X} \perp\!\!\!\perp_{\mathcal{G}} \boldsymbol{Y} \mid \boldsymbol{Z} \Leftrightarrow \boldsymbol{X} \perp\!\!\!\perp \boldsymbol{Y} \mid \boldsymbol{Z}$.

Given two vertices $X$ and $Y$ in $\mathcal{G}$, we use $X \prec Y$ to denote that $X$ is an ancestor of $Y$. Equivalently, $Y \succ X$ denotes that $Y$ is a descendant of $X$. We write $X \preceq Y$ if $X$ is *not a descendant* of $Y$, in which case $X$ may or may not be an ancestor of $Y$. (Note that $X \not\prec Y$ implies $Y \preceq X$.) We write $X \sim Y$ when neither variable is an ancestor of the other, i.e. $X \not\prec Y$ and $Y \not\prec X$. In acyclic graphs, the ancestry relation imposes a *strict partial order* characterized by the following three properties:

- *Irreflexivity*: $X \prec X \vdash \text{FALSE}$.
- *Asymmetry*: $X \prec Y \vdash Y \not\prec X$.
- *Transitivity*: $X \prec Y \wedge Y \prec Z \vdash X \prec Z$.

Relations $\prec$ and $\preceq$ can be applied to pairs of sets, implying that the relation holds between each pair of elements from the Cartesian product of the respective sets.

Let $\mathbb{E}[Y \mid do(X = x)]$ denote the expected *outcome* $Y$ under an intervention that fixes the *treatment* $X$ to level $x$. *Covariate adjustment* postulates that

$$\mathbb{E}[Y \mid do(X = x)] = \int \mathbb{E}[Y \mid x, \boldsymbol{z}]\, p(\boldsymbol{z})\, d\boldsymbol{z}$$

for a set of vertices $\boldsymbol{Z}$ in $\mathcal{G}$. It holds if $\boldsymbol{Z}$ satisfies the *backdoor criterion* with respect to $(X, Y)$ [Pearl, 2009a, Ch. 3], in which case we say that $\boldsymbol{Z}$ is a *valid adjustment set* for $(X, Y)$. Complete graphical criteria for covariate adjustment can be found in [Shpitser et al., 2010, Perković et al., 2018].

Finally, we define minimal (de)activators, originally highlighted by Claassen and Heskes [2011]:

**Definition 1** (Minimal activator)**.** Variable $D$ is a *minimal activator* of the relationship between $A$ and $B$ given $\boldsymbol{C}$ iff: (1) $A \not\!\perp\!\!\!\perp B \mid \boldsymbol{C} \cup D$; and (2) $A \perp\!\!\!\perp B \mid \boldsymbol{C}$. In this case, we write $A \not\!\perp\!\!\!\perp B \mid \boldsymbol{C} \cup [D]$. □

**Definition 2** (Minimal deactivator)**.** Variable $D$ is a *minimal deactivator* of the relationship between $A$ and $B$ given $\boldsymbol{C}$ iff: (1) $A \perp\!\!\!\perp B \mid \boldsymbol{C} \cup D$; and (2) $A \not\!\perp\!\!\!\perp B \mid \boldsymbol{C}$. In this case, we write $A \perp\!\!\!\perp B \mid \boldsymbol{C} \cup [D]$. □

## 3  PROBLEM STATEMENT

Assume a DAG $\mathcal{G}$ contains observable vertices $\boldsymbol{Z} \cup \boldsymbol{X}$, consisting of background variables $\boldsymbol{Z}$ and foreground variables $\boldsymbol{X}$. Let $|\boldsymbol{Z}| = d_Z$ and $|\boldsymbol{X}| = d_X$, with potentially $d_Z \gg d_X$. $\mathcal{G}$ may have unobserved vertices $\boldsymbol{U}$ with more than one descendant in $\boldsymbol{Z} \cup \boldsymbol{X}$ (i.e., unmeasured confounders). We construct the latent projection of $\mathcal{G}$ by marginalizing over hidden variables, replacing any path $X_i \leftarrow U_{ij} \rightarrow X_j$ with a bidirected edge $X_i \leftrightarrow X_j$ to form an acyclic directed mixed graph (ADMG) with vertex set $\boldsymbol{Z} \cup \boldsymbol{X}$. The symbol $\mathcal{G}^{\backslash \boldsymbol{U}}$ will denote the ADMG of $\mathcal{G}$.

The goal is to infer as much as possible about the causal structure of $\mathcal{G}_X \subset \mathcal{G}$, which consists of vertices $\boldsymbol{X}$ and the edges with endpoints in $\boldsymbol{X}$. We make the following assumptions:

(A1) $\mathcal{G}$ is acyclic.
(A2) $p(\boldsymbol{z}, \boldsymbol{x})$ is faithful to $\mathcal{G}^{\setminus U}$.
(A3) $\boldsymbol{Z}$ contains no descendant of $\boldsymbol{X}$ in $\mathcal{G}$, i.e. $\boldsymbol{Z} \preceq \boldsymbol{X}$.

The first assumption can be relaxed—$\mathcal{G}_Z$ may contain cycles under some conditions—but we adopt it here to avoid further technicalities. Faithfulness is a common yet somewhat controversial starting point for many causal discovery procedures (more on this in Sect. 7). The ordering assumption applies in many settings where background knowledge permits a categorical distinction between upstream and downstream variables, e.g. when data are recorded at different times.

For any pair of variables $X, Y \in \boldsymbol{X}$, exactly one of three possibilities obtains: (G1) $X \prec Y$; (G2) $X \succ Y$; or (G3) $X \sim Y$. Our discovery problem is defined as deciding which relationship holds for each pair of vertices in $\mathcal{G}_X$. A similar goal motivates Magliacane et al. [2016], who derive a general algorithm called ancestral causal inference (ACI). ACI does not exploit the the background-foreground split and does not scale to high dimensionality. It also comes with no theory about the error control of its inferences. Note that some relationships in $\mathcal{G}_X$ may be only partially identifiable, e.g. if all we can determine is that $X \preceq Y$. Others may be entirely unidentifiable, e.g. if latent confounding is present.

In the next section, we describe a causal discovery algorithm that assumes we have an *oracle* capable of returning exact information about which conditional independencies hold in the population. This is so that we can more easily discuss the limits of what can in principle be discovered from the assumptions provided. In Sect. 5, we present a practical statistical algorithm with error control guarantees.

# 4 CONFOUNDER BLANKETS AND THE ORACLE ALGORITHM

There are sound and complete procedures, based on the fast causal inference (FCI) algorithm of Spirtes et al. [2000], which return all and only those graphs that are Markov equivalent to the true $\mathcal{G}$ [Zhang, 2008].[1] Such methods scale poorly with data dimensionality, as they must query for conditional independence over an exponentially increasing number of candidate conditioning sets. For tractability, sometimes it is assumed that $\mathcal{G}$ is sparse or small [e.g., Magliacane et al., 2016], an unrealistic assumption if we

---

[1] FCI returns a partial ancestral graph (PAG), an equivalence class of maximal ancestral graphs (MAGs) [Richardson and Spirtes, 2002]. By contrast, the PC algorithm, which assumes causal sufficiency, returns a completed partially directed acyclic graph (CPDAG), an equivalence class of DAGs [Spirtes et al., 2000].

think each element of $\boldsymbol{X}$ should be directly connected to a substantive fraction $\mathcal{O}(d_Z)$ of background variables—a type of structure taken for granted in most methods that estimate causal effects by covariate adjustment [Hernán and Robins, 2009]. Instead, this work is based on the following principle:

**Definition 3** (The Confounder Blanket Principle, CBP). In the presence of a large set of background variables $\boldsymbol{Z}$, where it is believed that each element of $\boldsymbol{X}$ may be adjacent to $\mathcal{O}(d_Z)$ elements of $\boldsymbol{Z}$ in $\mathcal{G}^{\setminus U}$, do not attempt to test for conditional independencies using arbitrary subsets of $\boldsymbol{Z}$. In particular, work under the expectation that if some $\boldsymbol{A} \subset \boldsymbol{Z} \cup \boldsymbol{X}$ is a valid adjustment set for any ordered pair $X_i \prec X_j$, then $\boldsymbol{A} \cup \boldsymbol{Z}$ is also valid. We call a set of background variables with this property a *confounder blanket*. □

A failure of CBP does not compromise the *soundness* of the algorithms presented in the sequel, but it may affect their *completeness*. In particular, under CBP, we are exposed to the problem of *M-structures* [Pearl, 2009b], where some $Z \in \boldsymbol{Z}$ is a collider on a path $X_i \leftarrow \cdots \rightarrow Z \leftarrow \cdots \rightarrow X_j$. Without searching through subsets of $\boldsymbol{Z}$, it may be difficult or impossible to infer the causal order of $\mathcal{G}_X$ in this setting.

M-structures can indeed make a substantive impact to the bias of an adjustment set. However, Ding and Miratrix [2015] have shown that, at least at under some reasonable distributions of parameters in some parametric models, their impact may be negligible with high probability, and hence, statistically hard to detect in a causal discovery method. Instead of proposing yet another derivative of FCI, we believe that practitioners with access to a large set of background variables—which may be required in order to stand a chance against unmeasured confounding—are better served by methods grounded in the CBP.

## 4.1 STRUCTURAL SIGNATURES AND ALGORITHM

Our algorithm will be based on the following inference rules, adapted from Entner et al. [2013] and Magliacane et al. [2016]. In what follows, let $\boldsymbol{A}$ and $\{X, Y\}$ be two sets of observable vertices in a DAG $\mathcal{G}$, where $\boldsymbol{A} \preceq \{X, Y\}$, and let $\boldsymbol{A}_{\setminus W} := \boldsymbol{A} \setminus \{W\}$ for some vertex $W$. Our first rule detects (indirect) causes via relations of minimal independence:

(R1) If $\exists W \in \boldsymbol{A} : W \perp\!\!\!\perp Y \mid \boldsymbol{A}_{\setminus W} \cup [X]$, then $X \prec Y$.

The soundness of (R1), and (R2) below, follows immediately from Lemma 1 of Magliacane et al. [2016], combined with the partial order $\boldsymbol{A} \preceq \{X, Y\}$. (R1) applies when $X$ *deactivates* all paths from $W$ to $Y$. When this structure obtains, causal effects can be estimated using the backdoor adjustment with admissible sets $\boldsymbol{A}$ and $\boldsymbol{A}_{\setminus W}$.

Our second inference rule eliminates (indirect) causes via

**Algorithm 1** CBL-ORACLE

**Input:** Background set $\boldsymbol{Z}$, foreground set $\boldsymbol{X}$, oracle $\mathcal{I}$
**Output:** Ancestrality matrix $\mathbf{M}$

Initialize: converged $\leftarrow$ FALSE, $\mathbf{M} \leftarrow [\texttt{NA}]$
**while not** converged **do**
  converged $\leftarrow$ TRUE
  **for** $X_i, X_j \in \boldsymbol{X}$ such that $i > j$, $\mathbf{M}_{ij} = [\texttt{NA}]$ **do**
    $\boldsymbol{A} \leftarrow \boldsymbol{Z} \cup \big\{ X \in \boldsymbol{X} \backslash \{X_i, X_j\} : X \preceq_{\mathbf{M}} \{X_i, X_j\} \big\}$
    **if** $\mathcal{I}(X_i \perp\!\!\!\perp X_j \mid \boldsymbol{A})$ **then**
      $\mathbf{M}_{ij} \leftarrow i \sim j$, converged $\leftarrow$ FALSE
    **else**
      **for** $W \in \boldsymbol{A}$ **do**
        **if** $\mathcal{I}(W \perp\!\!\!\perp X_j \mid \boldsymbol{A}_{\backslash W} \cup [X_i])$ **then**
          $\mathbf{M}_{ij} \leftarrow i \prec j$, converged $\leftarrow$ FALSE
        **else if** $\mathcal{I}(W \perp\!\!\!\perp X_i \mid \boldsymbol{A}_{\backslash W} \cup [X_j])$ **then**
          $\mathbf{M}_{ij} \leftarrow j \prec i$, converged $\leftarrow$ FALSE
        **else if** $\mathcal{I}(W \not\perp\!\!\!\perp X_j \mid \boldsymbol{A}_{\backslash W} \cup [X_i])$ **then**
          $\mathbf{M}_{ij} \leftarrow \mathbf{M}_{ij} \wedge j \preceq i$, converged $\leftarrow$ FALSE
        **else if** $\mathcal{I}(W \not\perp\!\!\!\perp X_i \mid \boldsymbol{A}_{\backslash W} \cup [X_j])$ **then**
          $\mathbf{M}_{ij} \leftarrow \mathbf{M}_{ij} \wedge i \preceq j$, converged $\leftarrow$ FALSE
        **end if**
      **end for**
    **end if**
  **end for**
  $\mathbf{M} \leftarrow$ CLOSURE($\mathbf{M}$)
**end while**

relations of minimal dependence:

(R2)  If $\exists W \in \boldsymbol{A} : W \not\perp\!\!\!\perp X \mid \boldsymbol{A}_{\backslash W} \cup [Y]$, then $X \preceq Y$.

(R2) applies when $Y$ *activates* some path from $W$ to $X$. This means that $Y$ must be a (descendant of a) collider on that path, and cannot be a non-collider on any other path active under $\boldsymbol{A}_{\backslash W}$.

Our third rule establishes causal independence via separating sets, and follows immediately from faithfulness:

(R3)  If $X \perp\!\!\!\perp Y \mid \boldsymbol{A}$, then $X \sim Y$.

These building blocks are the basis for our confounder blanket learner (CBL), outlined in Alg. 1. CBL-ORACLE outputs a square, lower triangular ancestrality matrix $\mathbf{M}$, with $\mathbf{M}_{ij}$ representing the partial order between vertices $(X_i, X_j)$. The subscript $\mathbf{M}$ on a partial ordering relation indicates that it is already encoded in the matrix, which evolves with each pass through the for loop. The oracle $\mathcal{I}$ is an indicator function over conditional independencies on $p(\boldsymbol{z}, \boldsymbol{x})$. Note that inferences derived via (R2) are recorded as conjuncts, since they are consistent with multiple structures. The CLOSURE operation, fully articulated in Appx. A, ensures that $\mathbf{M}$ satisfies the characteristic properties of a strict partial order, thereby reducing conjunctions to their most informative implication.

## 4.2 PROPERTIES

Proofs for all theorems are given in Appx. A.

**Theorem 1** (Soundness). *All ancestral relationships returned by* CBL-ORACLE *hold in the true* $\mathcal{G}_X$. *Moreover, if* $\mathbf{M}_{ij} = i \prec j$, *then the set of shared non-descendants* $\boldsymbol{A} = \boldsymbol{Z} \cup \big\{ X \in \boldsymbol{X} \backslash \{X_i, X_j\} : X \preceq_{\mathbf{M}} \{X_i, X_j\} \big\}$ *is a valid adjustment set for* $(X_i, X_j)$.

By design, CBL-ORACLE can be uninformative where a method like FCI will provide a causal order. One of the simplest examples is the so-called *Y-structure* [Mani et al., 2006], $\{X_1 \rightarrow X_3, X_2 \rightarrow X_3, X_3 \rightarrow X_4\}$, where FCI discovers $X_3 \rightarrow X_4$. By contrast, with an empty $\boldsymbol{Z}$, CBL-ORACLE cannot infer any causes (though it may still infer $X_1 \sim X_2$ via (R3)). However, the presence of a single edge from a background variable $Z$ into $X_1$, $X_2$, or $X_3$ will allow for the discovery that $X_3 \prec X_4$, while an edge from $Z$ into $X_4$ will allow for the discovery that $X_3 \preceq X_4$.

We characterize full identifiability conditions for Alg. 1 as follows, with $\boldsymbol{X}_{\preceq i}$ standing for the set of all $X_i$'s observable non-descendants in $\mathcal{G}$, including $X_i$ itself. Without loss of generality, assume that sets are indexed such that no $X_i$ is a descendant of some $X_j$ for $j > i$.

**Theorem 2** (Identifiability). *The following conditions are sufficient for* CBL-ORACLE *to learn the total causal order of* $\mathcal{G}_X$. *If* $X_i \sim X_j$, *then either (i) there is no active backdoor path between* $X_i$ *and* $X_j$ *given* $\boldsymbol{Z}$ *and their common ancestors in* $\boldsymbol{X}$; *or (ii) some* $V_i \in \boldsymbol{X}_{\preceq i}$ *is d-connected to* $X_j$ *given* $\boldsymbol{X}_{\preceq i} \backslash \{V_i\}$, *and some* $V_j \in \boldsymbol{X}_{\preceq j}$ *is d-connected to* $X_i$ *given* $\boldsymbol{X}_{\preceq j} \backslash \{V_j\}$. *If* $X_i \prec X_j$, *then either (iii) some* $V \in \boldsymbol{X}_{\preceq i}$ *is d-separated from* $X_j$ *given* $\boldsymbol{X}_{\preceq i} \backslash \{V\}$; *or (iv) there exists a nonempty set of mediators along a unidirectional path* $X_i \prec X_{i+1} \prec \cdots \prec X_{j-1} \prec X_j$ *such that condition (iii) applies to each pair* $\{X_k, X_{k+1}\}, k \in \{i, \ldots, j-1\}$.

Condition (i) above motivates the name *confounder blanket*.

One of the key points, however, is the *completeness* of our algorithm. In the nonparametric causal discovery literature, this is usually defined with respect to an oracle that delivers true answers to all conditional independence queries over observable variables. We define a new scope for completeness that places some reasonable limits on oracular omnipotence. First, we introduce the following definitions:

**Definition 4** (Iteration-$t$ known non-descendant). Given an algorithm $\mathcal{A}$, we call vertex $W$ an *iteration-$t$ known non-descendant* of a vertex $X$ if either (i) $W \in \boldsymbol{Z}$; or (ii) after $t$ modifications to $\mathbf{M}$ by $\mathcal{A}$, we have $W \preceq_{\mathbf{M}} X$. $\square$

**Definition 5** (Lazy oracle algorithm). Let $\boldsymbol{X}_{\preceq i}^t$ be the set of all iteration-$t$ known non-descendants of $X_i$ according to some algorithm $\mathcal{A}$. A *lazy oracle algorithm* is one that starts with an uninformative ancestrality matrix $\mathbf{M}$ and updates at each round $t$ with answers to queries of just two types:

(i) $W \perp\!\!\!\perp X_i \mid \boldsymbol{S}_{ij \setminus W}^t \cup \phi(X_j)$, such that $W \in \boldsymbol{S}_{ij}^t$ and $\phi(X_j) \in \{\emptyset, \{X_j\}\}$; and

(ii) $X_i \perp\!\!\!\perp X_j \mid \boldsymbol{S}_{ij}^t$,

where $\{X_i, X_j\} \subseteq \boldsymbol{X}$ and $\boldsymbol{S}_{ij}^t := \boldsymbol{X}_{\preceq i}^t \cap \boldsymbol{X}_{\preceq j}^t$. $\square$

Our oracle may be clairvoyant when it comes to probabilistic relationships, but she is not quite as accommodating as her classical counterpart. In particular, she refuses to marshal her powers in service of combinatorial search strategies, which she considers tedious and inelegant. Instead she bestows her favor upon us only when we limit ourselves to a more restrictive class of queries pertaining to independence relationships conditioned on the complete set of (known) non-descendants for any given pair of foreground variables.

Observe that inferences about ancestral relationships are fully ordered with respect to their information content: $\{\text{NA}\} \prec \{i \preceq j\} \prec \{i \prec j\} \sim \{i \sim j\}$. This motivates the following optimality target:

**Definition 6** (Dominance). Among the set of all sound procedures for learning ancestral relationships, we say that algorithm $\mathcal{A}$ *dominates* algorithm $\mathcal{B}$ iff $\mathcal{A}$ is strictly more informative than $\mathcal{B}$. That is, (i) there exists no pair of observable vertices in any DAG $\mathcal{G}$ such that $\mathcal{A}$'s output for that pair is less informative than $\mathcal{B}$'s; and (ii) there exists some pair of observable vertices in some DAG $\mathcal{G}$ such that $\mathcal{A}$'s output for that pair is more informative than $\mathcal{B}$'s. $\square$

Finally, we may state our completeness result.

**Theorem 3** (Completeness). *No lazy oracle algorithm dominates* CBL-ORACLE. *That is, inferences returned by* CBL-ORACLE *are always at least as informative as those of any lazy oracle algorithm.*

An immediate corollary of Thm. 3 is that the identifiability conditions of Thm. 2 are not just sufficient but also *necessary* with respect to a lazy oracle algorithm.

Of course, relationships of conditional independence are estimated from finite samples in practice. In the sequel, we consider practical methods for implementing an algorithm that is pointwise consistent under further assumptions about the nature of conditional independencies in $p(\boldsymbol{z}, \boldsymbol{x})$.

## 5   STATISTICAL INFERENCE

In this section, we describe a practical method based on the oracle algorithm, called CBL-SAMPLE, or simply CBL. Our main assumption to help bridge the gap between theory and practice is the following:

(A4) We have access to a regression algorithm by which we can test any pairwise conditional independence statement $X \perp\!\!\!\perp Y \mid \boldsymbol{S}$ by regressing $Y$ on $\boldsymbol{S} \cup \{X\}$.

The regression is implemented with a variable selection strategy which will, in the limit of infinite data, remove $X$ from the regression equation iff $X \perp\!\!\!\perp Y \mid \boldsymbol{S}$.

Statistical error control techniques are presented under this assumption. We will not discuss its validity for the specific sparse regression engines exploited here. This is well-understood in, for example, the case of Gaussian linear regression and a $z$-test of the coefficient for $X$. In other scenarios, due to computational or statistical reasons, this is less straightforward (e.g., lasso is "sparsistent" only under restrictive assumptions, and it is possible to have a covariate dropping out of a population regression function even if the corresponding conditional independence does not hold [Hastie et al., 2015]). Instead, we take this foundational assumption as an idealization that simplifies analysis, being open about the fact that, in practice, such assumptions may only be approximately satisfied.

Constraints like $W \perp\!\!\!\perp X_j \mid \boldsymbol{A}_{\setminus W} \cup [X_i]$ suggest two regression models per triplet $(X_i, X_j, W)$: one for the regression of $X_j$ on $W$, $\boldsymbol{A}_{\setminus W}$ and $X_i$, and another for the regression of $X_j$ on $W$ and $\boldsymbol{A}_{\setminus W}$ only. In the algorithm that follows, we simplify this by using a single model to simultaneously test for all $W$, fitting a regression for $X_j$ on $\boldsymbol{A}$ and $X_i$, and another regression for $X_j$ on $\boldsymbol{A}$ only. These are clearly mathematically equivalent (as $\boldsymbol{A} = \boldsymbol{A}_{\setminus W} \cup \{W\}$), so long as the variable selection procedure in the regression model can be computed exactly, for instance when using $z$-tests for a Gaussian regression model or when lasso sparsistency conditions are satisfied. This will not necessarily be the case when an intractable combinatorial search underlies variable selection, or when conditions for a continuous relaxation do not hold. The safer alternative is, just like in the oracle algorithm, to perform individualized model selection for each $W$, without any concern for simultaneously selecting variables within $\boldsymbol{A}_{\setminus W}$.

Nevertheless, for simplicity we rely on a joint variable selection procedure that uses all elements of $\boldsymbol{A}$ when fitting each regression model, and empirically show that bundling individual covariate tests achieves better results than existing alternatives. We emphasize that combinatorial search can be avoided altogether by separating selection on each $W$ from any sort of sparse regularization or search among the other covariates, if so desired.

**Bipartite Subgraphs.**   We begin with the simplest case, in which we have just two foreground variables $\boldsymbol{X} = \{X, Y\}$. We fit a quartet of models to estimate the following conditional expectations:

$$f_Y^0 : \mathbb{E}[Y \mid \boldsymbol{Z}] \qquad f_X^0 : \mathbb{E}[X \mid \boldsymbol{Z}]$$
$$f_Y^1 : \mathbb{E}[Y \mid \boldsymbol{Z}, X] \quad f_X^1 : \mathbb{E}[X \mid \boldsymbol{Z}, Y],$$

where subscripts index outcome variables and superscripts differentiate between full and reduced conditioning sets.

Assume, for concreteness, that all structural equations are linear. Since some elements of $\boldsymbol{Z}$ may not influence $\boldsymbol{X}$, we estimate the members of this quartet using lasso regression, which performs automatic feature selection. This results in four different *active sets* of predictors. For instance, the active set $\hat{\boldsymbol{S}}_Y^0(\lambda) \subseteq \boldsymbol{Z}$ picks out just those background variables that receive nonzero weight in the model $\hat{f}_Y^0$ at a given value of the regularization parameter $\lambda$ (though we generally suppress the dependence for notational convenience).

Our basic strategy is to refit the model quartet some large number of times $B$, taking different training/validation splits to get a sampling distribution over active sets. (The exact resampling method is described in more detail below.) This allows us to test the antecedent of (R3) by evaluating whether $X \in \hat{\boldsymbol{S}}_Y^1$ and $Y \in \hat{\boldsymbol{S}}_X^1$ with sufficient frequency. If the conjunction occurs fewer than $\gamma B$ times (with the convention that $\gamma = {}^1\!/{}_2$) we conclude that $X \sim Y$. Because we seek to minimize errors of commission, we are more conservative in our inference procedures for $\prec$ and $\preceq$ relations. From our distribution of active sets we calculate the (de)activation rate of each non-descendant with respect to a given causal ordering. This gives four unique rates per non-descendant, representing the (de)activation frequencies when treating either $X$ or $Y$ as the candidate cause. High rates are evidence that the corresponding inference rule applies.

What is a reasonable threshold for drawing such an inference? It is not immediately obvious how to specify an expected null (de)activation rate without further assumptions on the data generating process. Rather than introduce some ad-hoc prior or sparsity constraint, we take an adaptive approach inspired by the stability selection procedure of Meinshausen and Bühlmann [2010]. Specifically, we use a variant of complementary pairs stability selection [Shah and Samworth, 2013], which guarantees an upper bound on the probability of falsely selecting a low-rate feature at any given threshold $\tau$. The method is so named because, on each draw $b$, we partition the data into disjoint sets of equal size. Rates are estimated over all $2B$ subsamples.

Stability selection was originally conceived for controlling error rates in feature selection problems, primarily lasso regression. We adapt the procedure to accommodate our modified target, which is a conjunction of inclusion/exclusion statements rather than a single selection event. Specifically, we are interested in the probability of (de)activation under some fixed feature selection procedure $\hat{\boldsymbol{S}}$. We write:

$$r_d(Z)_{X \preceq Y} := \mathbb{P}(Z \in \hat{\boldsymbol{S}}_Y^0 \wedge Z \notin \hat{\boldsymbol{S}}_Y^1)$$

to denote the probability that feature $Z$ is deactivated w.r.t. $X \preceq Y$. Activation rates are analogously defined:

$$r_a(Z)_{X \preceq Y} := \mathbb{P}(Z \notin \hat{S}_X^0 \wedge Z \in \hat{S}_X^1).$$

For the opposite ordering, we simply swap active sets, using $\hat{\boldsymbol{S}}_X^0, \hat{\boldsymbol{S}}_X^1$ for deactivation, and $\hat{\boldsymbol{S}}_Y^0, \hat{\boldsymbol{S}}_Y^1$ for activation w.r.t. $X \succeq Y$.

**Definition 7** (Complementary pairs stability selection). Let $\{(\mathcal{D}_{2b-1}, \mathcal{D}_{2b}) \subseteq [n] : b \in [B]\}$ be randomly chosen independent pairs of sample subsets of size $\lfloor n/2 \rfloor$ such that $\mathcal{D}_{2b-1} \cap \mathcal{D}_{2b} = \{\emptyset\}$. For $\tau \in [0,1]$, $\phi \in \{a, d\}$, and $\psi \in \{X \preceq Y, X \succeq Y\}$, the complementary pairs stability selection (CPSS) procedure is $\hat{\boldsymbol{H}}_{\tau,\phi,\psi} := \{k : \hat{r}_\phi(Z_k)_\psi \geq \tau\}$, with estimated rates given by:

$$\hat{r}_d(Z)_{X \preceq Y} := \#\{b : Z \in \hat{\boldsymbol{S}}_Y^0(\mathcal{D}_b) \wedge Z \notin \hat{\boldsymbol{S}}_Y^1(\mathcal{D}_b)\}/2B$$

for deactivation w.r.t. $X \preceq Y$, and

$$\hat{r}_a(Z)_{X \preceq Y} := \#\{b : Z \notin \hat{\boldsymbol{S}}_X^0(\mathcal{D}_b) \wedge Z \in \hat{\boldsymbol{S}}_X^1(\mathcal{D}_b)\}/2B$$

for activation w.r.t. the same ordering. Again, to estimate rates for the opposite ordering, we simply swap active sets as described above. $\square$

For some $\theta < \tau$, let $\boldsymbol{L}_{\theta,\phi,\psi} := \{k : r_\phi(Z_k)_\psi \leq \theta\}$ denote the set of low-rate variable indices for some $\phi, \psi$. Our goal is to bound the expected number of low-rate features selected at a given threshold $\tau$, i.e. $\mathbb{E}[|\hat{\boldsymbol{H}}_{\tau,\phi,\psi} \cap \boldsymbol{L}_{\theta,\phi,\psi}|]$. Methods for doing so rely on certain assumptions about the distribution of rates for features within $\boldsymbol{L}_{\theta,\phi,\psi}$. Shah and Samworth [2013]'s tightest bound is achieved under $r$-concavity, formally defined in Appx. B. Roughly, $r$-concave distributions describe a continuum of constraints that interpolate between unimodality and log-concavity for $r \in [-\infty, 0]$. Simulation results suggest that (de)activation rates for low-rate features exhibit the following property (see Appx. B):

(A5) For all $Z \in \boldsymbol{L}_{\theta,\phi,\psi}$, empirical rates $\hat{r}_\phi(Z)_\psi$ are approximately $-1/4$-concave.

We now have the following error control guarantee:

**Theorem 4** (Error control). *The expected number of low-rate features selected by the CPSS procedure is bounded from above:*

$$\mathbb{E}[|\hat{\boldsymbol{H}}_{\tau,\phi,\psi} \cap \boldsymbol{L}_{\theta,\phi,\psi}|] \leq \min\{D(\theta^2, 2\tau - 1, B, -1/2),$$
$$D(\theta, \tau, 2B, -1/4)\}|\boldsymbol{L}_{\theta,\phi,\psi}|,$$

*where $D(\theta, \tau, B, r)$ is the maximum of $\mathbb{P}(X \geq \tau)$ over all $r$-concave random variables supported on $\{0, {}^1\!/{}_{2B}, {}^1\!/{}_B, \ldots 1\}$ with $\mathbb{E}[X] \leq \theta$.*

This is a direct application of Shah and Samworth [2013]'s Eq. 8. Though the bound is valid for all $\tau \in (\theta, 1]$, we apply an adaptive lower bound $\epsilon > \theta$, which denotes the minimum rate such that no conflicting inferences emerge, e.g. different ancestors deactivating for opposite causal orderings (see Alg. 5, Appx. C). We follow the authors' recommendations for default values of $B$ and $\theta$ (see Appx. B). We note that there is no closed form solution for $D(\theta, \tau, B, r)$, but the quantity is easily computed with numerical methods. If the number of (de)activations detected via this procedure exceeds the

maximum error bound of Thm. 4, we infer that at least one must be a true positive. Since even a single (de)activation is sufficient to partially order variable pairs, this licenses the corresponding inference.

**General case.** Our method can be expanded to accommodate larger sets of foreground variables and nonlinear structural equations. When $d_X > 2$, we simply loop through all $d_X(d_X-1)/2$ unique pairs of variables and record any inferences made at time $t = 1$. Like in the oracle algorithm, as the set $\boldsymbol{A}$ grows for $t > 1$, we continue cycling through pairs that have yet to be unambiguously decided until no further inferences are forthcoming. Though we use lasso regression for linear systems in our experiments, stepwise regression or even best subset selection may be viable alternatives [Hastie et al., 2020]. For nonlinear systems, we use gradient boosted regression trees with early stopping, which automatically adapt to signal sparsity [Friedman, 2001, Bühlmann and Yu, 2003]. Any function $s : \mathbb{R}^d \times \mathbb{R} \mapsto 2^d$ from input variables and outcome to an active set of predictors will suffice. Such feature selection subroutines may be consistent estimators for the Markov blanket of a given variable under fairly minimal conditions [see, e.g., Candès et al., 2018, Prop. 1].

In the worst case, CBL requires $\mathcal{O}(Bd_X^3)$ operations per feature selection subroutine $s$, the complexity of which itself presumably depends on $n, d_Z$ and $d_X$. For example, with $n > d = d_Z + d_X$, the least angle regression implementation of lasso takes $\mathcal{O}(d^3 + nd^2)$ computations [Efron et al., 2004], resulting in overall complexity of $\mathcal{O}\big(B(d_X^6 + nd_X^5 + d_Z^3)\big)$. More generally, if $s$ executes in polynomial time, then CBL is of complexity order P. Since constraint-based graphical learning without sparsity restrictions is NP-hard [Chickering et al., 2004], this represents a major computational improvement. The procedure can be further sped up by parallelizing over subsamples, as these are independent. For pseudocode summarizing CBL-SAMPLE, see Alg. 4 in Appx. C.

# 6 EXPERIMENTS

Full details of our simulation experiments are described in Appx. D. Briefly, we vary the sample size and dimensionality of the data, as well as graph structure and sparsity. Linear and nonlinear structural equations are applied at a range of different signal-to-noise ratios (SNRs).

**Bipartite subgraphs.** We benchmark against a constraint-based method proposed by Entner et al. [2013] and a score-based alternative similar in spirit to many causal discovery algorithms. We highlight two key differences between our proposal and Entner et al. [2013]'s: (1) Their method assumes a partial order on foreground variables upfront. With the prior knowledge that $X \preceq Y$, it tests whether $X \to Y$ or $X \sim Y$, with the possibility that the disjunction is undecidable from the observational distribution. It therefore has

an advantage in the following experiment, where the partial ordering assumption is satisfied, but competitors still consider the possibility that $X \leftarrow Y$. (2) The original version of Entner et al. [2013]'s method performs combinatorial search through the space of non-descendants, which is infeasible in our setting. Following the authors' advice, we simplify the procedure by sampling random variable-subset pairs from $\boldsymbol{Z}$, evaluating conditional independence either via partial correlation (for linear data) or the generalized covariance measure [Shah and Peters, 2020] with gradient boosting subroutine (for nonlinear data).

For our score-based benchmark, we train a series of models to evaluate three different structural hypotheses, corresponding to (G1) $X \to Y$; (G2) $X \leftarrow Y$; and (G3) $X \sim Y$. We use lasso for linear data and gradient boosting for nonlinear data. We calculate the proportion of variance explained on a test set for all settings. If (G3) scores highest, we return $X \sim Y$. Otherwise, we test whether out-of-sample residuals for the top scoring model are correlated with the foreground predictor. If so, we return NA; if not, we return whichever of (G1) or (G2) scored highest.

We visualize results for the setting with 100 background variables and expected sparsity $1/2$ (see Fig. 2). Data are simulated from 100 random graphs drawn under three different structural constraints: (a) $X \to Y$; (b) $X \perp\!\!\!\perp_{\mathcal{G}} Y \mid \boldsymbol{S}$, for some $\boldsymbol{S} \subseteq \boldsymbol{Z}$; and (c) $X \perp\!\!\!\perp_{\mathcal{G}} Y \mid \boldsymbol{S} \cup [\boldsymbol{U}]$, where $\boldsymbol{U}$ denotes a set of latent confounders. The first two are identifiable, while the third is not. Linear and nonlinear structural equations are applied with SNR = 2.

We find that CBL fares well in all settings. Constraint-based methods show less power to detect edges when present in this experiment, especially in nonlinear systems, while score-based methods incur higher error rates when edges are absent. We also observe that the constraint-based procedure requires considerable tuning—we had to experiment with a five-dimensional grid of decision thresholds to get reasonable results—and is by far the slowest to execute, taking about five times longer than CBL even with the random subset approach.

**Larger subgraphs.** We benchmark against two popular causal discovery algorithms: really fast causal inference (RFCI), a constraint-based method proposed by Colombo et al. [2012] as a more scalable version of the original FCI algorithm [Spirtes et al., 2000]; and greedy equivalence search (GES), a score-based alternative due to Meek [1997] and Chickering [2003]. Both algorithms can be computed with background information to encode our partial ordering assumption, and restricted to focus on the subgraph $\mathcal{G}_X$. Despite its name, RFCI struggles to converge in reasonable time ($< 24$ hours) when $n = 1000$ and $d_Z$ is on the order of 100, so we limit comparisons here to smaller datasets and run fewer replications for this method (5) than we do for GES (20). This illustrates how the assumption of extreme

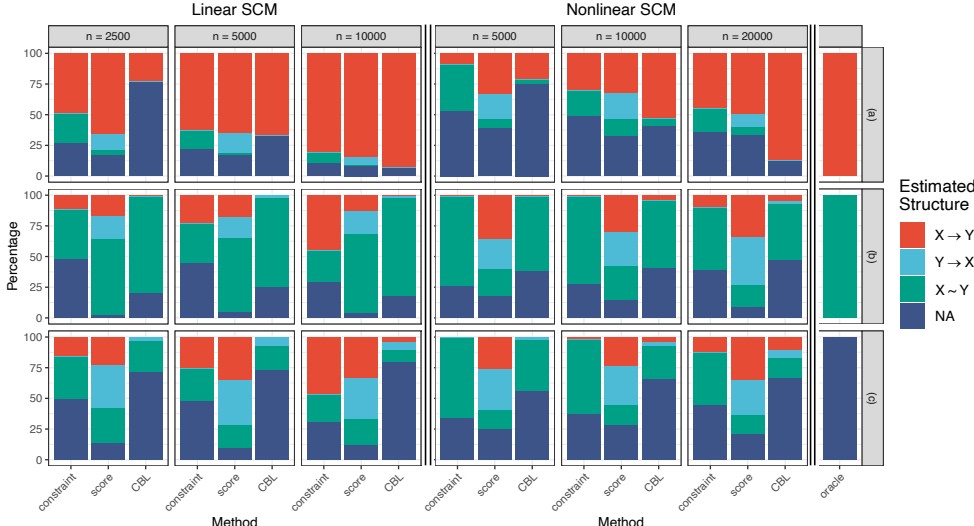

Figure 2: Simulation results at varying sample sizes for three different structures: (a) $X \rightarrow Y$; (b) $X \perp\!\!\!\perp_{\mathcal{G}} Y \mid \boldsymbol{S}$; and (c) $X \perp\!\!\!\perp_{\mathcal{G}} Y \mid \boldsymbol{S} \cup [\boldsymbol{U}]$. We compare our CBL method to constraint- and score-based benchmarks. Expected results of an independence oracle are included at the far right.

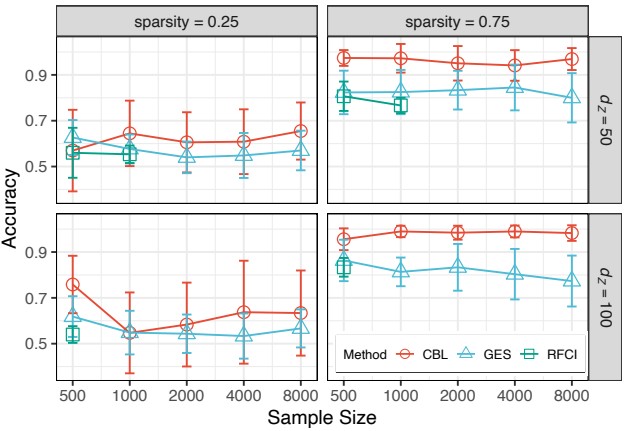

Figure 3: Simulation results for our multivariate experiment, benchmarking against RFCI and GES. Whiskers represent standard errors.

sparsity is necessary for RFCI to work in practice.

For this simulation, we draw random graphs of varying sample size with low (0.25) and high (0.75) sparsity, $d_Z \in \{50, 100\}$, and $d_X = 6$. Relationships are linear throughout, with RFCI using partial correlation tests for conditional independence and GES scoring edges according to BIC. Accuracy is measured with respect to all pairwise relationships for which a decision is reached. We find that CBL is more accurate on average in nearly all settings, with especially strong results in the high-sparsity, high-dimensionality regime. However, our method can be less stable than GES, as illustrated by the greater variance of results, particularly in dense networks where CBL outputs a relatively large number of NAs.

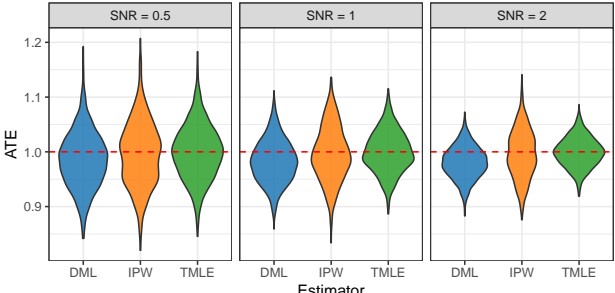

Figure 4: Average treatment effects estimated by combining CBL with three different algorithms at varying SNRs.

**Causal effects.** Since our method identifies admissible sets for all detected edges, we may estimate the average treatment effect (ATE) via backdoor adjustment. For this experiment, we simulate data from a partially linear model as originally parametrized by Robinson [1988]:

$$X = f(\boldsymbol{Z}) + \epsilon_X, \qquad \mathbb{E}[\epsilon_X \mid \boldsymbol{Z}] = 0,$$
$$Y = \beta X + g(\boldsymbol{Z}) + \epsilon_Y, \quad \mathbb{E}[\epsilon_Y \mid \boldsymbol{Z}, X] = 0,$$

with $X \in \{0, 1\}$ and $Y \in \mathbb{R}$. The goal is to estimate $\beta$, which corresponds to the ATE. We run our pipeline with three different estimators: double machine learning (DML) [Chernozhukov et al., 2018], inverse propensity weighting (IPW) [Rosenbaum and Rubin, 1983], and targeted maximum likelihood estimation (TMLE) [van der Laan and Rose, 2011]. For all three methods, models are fit with gradient boosting and parameters estimated via cross-fitting to avoid regularization bias. We simulate 1000 datasets with $\beta = 1, d_Z = 100, n = 10000$, and SNR $\in \{1/2, 1, 2\}$. We find that all three methods provide consistent ATE es-

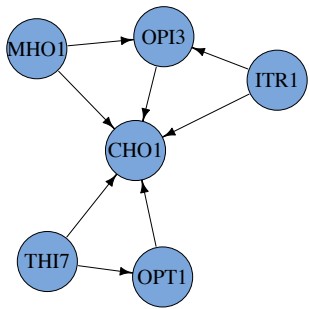

Figure 5: Estimated phosphocoline subnetwork in *S. cerevisiae*. Nodes are genes, edges denote ancestral relations.

timates, with TMLE generally performing best in terms of bias and variance (see Fig. 4). This illustrates how CBL can be combined with existing algorithms to go beyond causal discovery and into causal inference.

**Biological data.** As a final example, we consider regulatory mechanisms in *Saccharomyces cerevisiae*. Our background variables are single nucleotide polymorphism (SNP) markers, with transcriptomic profiles serving as foreground variables. Such setups are common in expression quantitative trait loci (eQTL) studies, where the goal is to identify genetic sources of variation in mRNA expression. Our dataset spans 112 $F_1$ segregants, a cross of parental strains BY4716 and the wild isolate RM11-1a [Brem and Kruglyak, 2005]. This dataset has been analyzed by several groups, who have identified numerous regulatory mechanisms through a combination of statistical and experimental methods [Brem et al., 2005, Storey et al., 2005, Chen et al., 2007].

We focus on the six genes that comprise the phosphocoline subnetwork, which regulates metabolic processes. The full set of background variables includes 3244 SNP markers, covering over 99% of the genome. We examine cis-eQTL candidates for each pair of genes—here defined as markers within 5 kilobases of either on the same chromosome—meaning $d_Z$ is usually on the order of 500. We use lasso for feature selection. Results are visualized in Fig. 5, where edges denote relations of ancestry rather than direct causation. These findings corroborate those of Chen et al. [2019], who recently examined regulatory mechanisms in yeast, and inferred a phosphocoline subgraph that includes each ancestral relationship depicted above. Our method is conservative by comparison, perhaps due to the acyclicity assumption. Chen et al. [2019] infer several cycles (e.g., between ITR1 and MHO1) where CBL withholds judgment.

# 7 DISCUSSION

We have proposed a novel method for learning ancestral relationships in downstream subgraphs based on the confounder blanket principle, which advises against combina-

torial search for conditioning sets in cases where scale-free sparsity cannot be safely assumed. Our CBL algorithm is provably sound and lazy oracle-complete. Our sample version controls errors of commission with high probability and compares favorably to constraint- and score-based alternatives in a range of trials. In addition to accurately learning ancestral relationships, CBL identifies valid adjustment sets for causal effect estimation.

Completeness in causal discovery has traditionally been defined with respect to a classical independence oracle. In the context of statistical inference, this idealization serves a clear purpose, since there exists no uniformly valid test of conditional independence [Robins et al., 2003, Shah and Peters, 2020]. Yet if our goal is simply to avoid the messiness of probabilistic reasoning in finite samples, then such oracles may overshoot the mark, for not only are they *omniscient* about conditional independence relations—they are also *omnipotent* with respect to computational complexity, able to scan through arbitrary subsets at no cost. We believe there are theoretical and practical advantages to decoupling these superpowers. Our lazy oracle is one example of how this may look, but others are also worth exploring.

We note several limitations of our method. First, CBL will struggle in the presence of weak edges. For instance, if the true graph is $Z \rightarrow X \rightarrow Y$ and $I(X;Y) \gg I(X;Z)$, then conditioning on $Y$ in finite samples could deactivate some path(s) from $Z$ to $X$, leading to the erroneous inference $Y \rightarrow X$. We observe that weak edges pose problems for all causal discovery procedures. Indeed, one motivation for taking an inclusive approach to background variables is the hope that a sufficiently large confounder blanket should include at least some strong edges that can be exploited to learn structural information about $\mathcal{G}_X$.

CBL relies on the faithfulness assumption, which has been challenged by numerous authors [Zhang and Spirtes, 2008, Andersen, 2013, Uhler et al., 2013]. Several weaker variants have been proposed, including SGS-minimality [Spirtes et al., 2000], P-minimality [Pearl, 2009a], and 2-adjacency faithfulness [Marx et al., 2021]. One direction for future work is to extend CBL under these relaxed assumptions.

The current implementation of CBL is order-dependent, insomuch as estimated subgraphs for the same dataset may vary if columns are reordered. This can be addressed using methods previously devised for constraint-based causal discovery [Colombo and Maathuis, 2014].

# Acknowledgements

This work was supported by ONR grant 62909-19-1-2096. We thank Joshua Loftus for helpful comments on an earlier draft of this manuscript, and Michael Barnes for fruitful discussions on eQTL analysis.

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
