# OpenReview forum: "Causal discovery under a confounder blanket"
_auai.org/UAI/2022/Conference — UAI 2022 Poster_

### Official Review · Reviewer_ziGt · 2022-03-25

**Q2(1) Originality/Novelty:** 3
**Q2(2) Significance/Impact:** 3
**Q2(3) Correctness/Technical Quality:** 3
**Q2(6) Clarity Of Writing:** 4
**Q6 Overall Score:** 7
**Q8 Confidence In Your Score:** 3

**Q1 Summary And Contributions:**

The authors propose a (as far as I know) novel way of inferring the causal relationships among a sub-graph given a set of confounding covariates (confounder blanket). An algorithm CBL-oracle is proposed and soundness, identifiability and completeness wrt. a "lazy oracle" are proven. Moreover, for the finite sample setting, statistical error control is provided using stability selection. Simulation studies empirically validate the proposed method.



**Q2 Assessment Of The Paper:**

More detailed information regarding each of these aspects is given below:

**Q2(4) Quality Of Experiments (Optional):**

3: Good: The experimental evaluation is adequate, and the results convincingly support the main claims.

**Q2(5) Reproducibility:**

3: Good: Key resources (e.g., proofs, code, data) are available and key details (e.g., proofs, experimental setup) are sufficiently well-described for competent researchers to confidently reproduce the main results.

**Q3 Main Strengths:**

- Very good quality of writing
- Result could have large impact in practice
- Clear theoretical results plus finite sample results

**Q4 Main Weakness:**

- complex notation, might be hard to digest without examples

**Q5 Detailed Comments To The Authors:**

I was impressed by the overall high quality of this paper. The main weakness (as indicated above) is from my point of view that the paper is quite densely written and contains. Thus, the key insights might be obscured, especially Def 5 (Lazy oracle algorithm) and Theorem 4 (Error control). I know space is limited, but I wonder if a (phps running) example might be used to illustrate the main points to fix ideas.

**Q7 Justification For Your Score:**

Paper seems strong in several dimensions; mainly readability could be improved as suggested above.

**Q9 Complying With Reviewing Instructions:**

1: Yes.

---

### Official Review · Reviewer_cFiD · 2022-04-07

**Q2(1) Originality/Novelty:** 3
**Q2(2) Significance/Impact:** 3
**Q2(3) Correctness/Technical Quality:** 3
**Q2(6) Clarity Of Writing:** 4
**Q6 Overall Score:** 6
**Q8 Confidence In Your Score:** 4

**Q1 Summary And Contributions:**

A method (CBL-ORACLE) is proposed for inferring causal relationships among a restricted (sub)system of variables when a large blanket of observed background confounders is available for deconfounding the target system. The method does not require making any sparsity assumption. The authors show the mehtod is sound and complete under a weak set of assumptions, and in the presence of latent confounding. Moreover, they provide finite sample guarantees given suitable variable selection methods.

**Q2 Assessment Of The Paper:**

More detailed information regarding each of these aspects is given below:

**Q2(4) Quality Of Experiments (Optional):**

2: Fair: The experimental evaluation is weak: important baselines are missing, or the results do not adequately support the main claims.

**Q2(5) Reproducibility:**

3: Good: Key resources (e.g., proofs, code, data) are available and key details (e.g., proofs, experimental setup) are sufficiently well-described for competent researchers to confidently reproduce the main results.

**Q3 Main Strengths:**

The work is well motivated with concrete examples (e.g., biomolecular systems with genetic data) of real-world settings in which CBL-ORACLE would be useful. The causal inference rules used in the method are not new, but the context in which they are used, as well as the accompanying proofs on soundness, completeness, and error control are novel, to the best of my knowledge.

The technical quality of the proposed work is high, as the authors even provide finite-sample guarantees for the performance of their CBL-ORACLE method. The key steps are sufficiently well described for reproducibility. As far as clarity is concerned, I would like to commend the authors for writing a virtually flawless paper, which was a pleasure to read.

**Q4 Main Weakness:**

The authors claim to have tested their approach "on simulated and real-world data", but I could only find examples of simulated data. I understand that real-world examples with known ground truth are difficult to find for this type of problems. Nevertheless, I would adjust the claim made in the introduction, since it seems misleading as currently written.

The authors neglect to discuss their claim of "polynomial time complexity" for CBL-ORACLE. I would think that the promised complexity needs to be shown either theoretically or empirically, at least in the supplementary material.

In the experimental evaluation, I would have expected comparisons against methods for Markov blanket induction (e.g., Aliferis et al., 2010). Finally, I believe some details are missing when it comes to reproducing the experiment where there are unobserved confounders. I could not find the relevant information on the latent confounders in the simulation, neither in the main paper, nor in the supplement.

**Q5 Detailed Comments To The Authors:**

On page 4, in the last sentence before subsection 4.3, I think it would be more useful to use the exact algorithm name specified in **Algorithm 1**, that is, CBL-ORACLE instead of "(CBL) oracle".

Following Definition 7, why does $\hat{\textbf{H}}$ have a hat symbol on it, while $\textbf{L}$ does not. These two sets seem to have equivalent definitions, so I would keep the notation consistent.

In Theorem 4, it is not clear to me what the intermediary values there are in the support set $\{0, 1/2B, 1/B, ... 1}$. What is the rule being followed when defining the set?

In Section 6.2, how do you define sparsity? What does 0.25 / 0.75 sparsity mean concretely? Does a higher number correspond to more sparsity?

In the Discussion section on page 8, you note that the " one motivation for taking an inclusive approach to background variables is the hope that a sufficiently large confounder blanket should include at least some strong edges that can be exploited to learn structural information about $\mathcal{G}_X$. How does that fit into your example of using genetic data as background variables. Individual genetic associations are typically quite weak and require hundreds of thousands of observations to be deemed significant. Would you be able to find any strong enough edges for CBL-ORACLE to work?

In the reference section (right column), I believe the last name of one of the authors has been duplicated ("Subramani Mani Mani").

**Q7 Justification For Your Score:**

The paper is especially strong when it comes to the writing (Q2.6) and technical quality (Q2.3), both of which weigh heavily in my score. I think the way in which a paper is written is key for its future dissemination, while at the same time, the work within needs to be technically sound. I also weighed the paper's potential significance (Q2.2) more heavily, because it is important for a different type of approach to causal discovery and inference to find application in real-world settings.

**Q9 Complying With Reviewing Instructions:**

1: Yes.

---

### Official Review · Reviewer_m4c9 · 2022-04-09

**Q2(1) Originality/Novelty:** 3
**Q2(2) Significance/Impact:** 3
**Q2(3) Correctness/Technical Quality:** 3
**Q2(6) Clarity Of Writing:** 4
**Q6 Overall Score:** 7
**Q8 Confidence In Your Score:** 3

**Q1 Summary And Contributions:**

This paper considers a causal discovery setting where the task is to learn the ancestral relationships between a set of foreground variables X that are causally preceded by a potentially large set of background variables Z. Guided by their "Confounder Blanket Principle", the authors introduce a sound and complete algorithm for this task. Numerical experiments that compare to constrained-based and score-based baseline algorithms show promising results.

**Q10 Ethical Concerns (Optional):**

No ethical concerns.

**Q2 Assessment Of The Paper:**

More detailed information regarding each of these aspects is given below:

**Q2(4) Quality Of Experiments (Optional):**

3: Good: The experimental evaluation is adequate, and the results convincingly support the main claims.

**Q2(5) Reproducibility:**

3: Good: Key resources (e.g., proofs, code, data) are available and key details (e.g., proofs, experimental setup) are sufficiently well-described for competent researchers to confidently reproduce the main results.

**Q3 Main Strengths:**

The paper is exceptionally well written, it is a pleasant read and very precise in most of its parts. The proposed "Confouder Blancket Principle, CBP" is well motivated and innovative. The proposed CBL-Algorithm is easy to understand in its basic setup and appears to be technically sound, although I did not check the proofs and did not go in the details of the algorithm's sample version that are presented in the appendix. Notably, the paper entails a thorough statistical analysis of the algorithm's sample version including an error contral guarantee. The numerical experiments show promising results for CBL in comparison to a few baseline algorithms. The CBP and CBL-Algorithm will, I think, inspire further research. Code in R is provided.

**Q4 Main Weakness:**

The numerical experiments are sufficient in my opinion and do support the paper's claim. However, more could be done: It would be interesting to see how CBL behaves for varing numers $d_Z$ of background variables, especially for small $d_Z$. There are few places in which explanations could be expanded, see Q5 below.

**Q5 Detailed Comments To The Authors:**

Major comments:

- Definition 7 and below: Looking at the definitions of  $\hat{\mathbf{H}}$ and $\mathbf{L}$, shouldn't their intersection always be empty due to $\theta < \tau$. Did you, perhaps, forget "hat" symbols on $r_{\phi}(Z_k)_{\psi}$ in Definition 7?

- "If the number of (de)activations detected via this method exceeds the maximum error bound of Thm. 4, we infer that at least one must be a true positive. Since even a single (de)activation is sufficient to partially order variable pairs, this licenses the corresponding inference": This is entirely unclear to me but, looking at Algorithm 2, seems to be essential. Please clarify this in the final version.


Minor comments:

- "Under a structural assumption that, we argue, must be satisfied in practice if informative answers are to be found": To me it did not become clear which of the assumptions you are referring to with this. This could be clarified in the final version.

- "The goal is to infer as much as possible about the causal structure of $\mathcal{G}_X \subset \mathcal{G}$": Should it be the ADMG $\mathcal{G}^{\setminus \mathbf{U}}$ instead of the DAG $\mathcal{G}$ here? The same in Theorem 1.

- Potential TeX issue regarding the equation labels (A1), (A2), (A3), (R1), (R2), (R3), (A4): These lables extend beyond the left border of the text.

- "The first assumption can be relaxed": Be more specific here. I guess you are saying it can be relaxed to $\mathcal{G}_X$ being acyclic.

- "If we succeed in learning which structure holds for each variable pair, then we recover the complete subgraph $\mathcal{G}_X$": I am not sure this entirely correct. Consider, for example, the following two DAGs: DAG G_1 is X_1 --> X_2 --> X_3. DAG G_2 is X_1 --> X_2 --> X_3 with X_1 --> X_3. These DAGs agree on the ancestral relationship between any pair of variables, but they are distinct.

- "There are sound and complete algorithms, based on the fast causal inference (FCI) algorithm ... which are equivalent to the true $\mathcal{G}$": There seems to be a slight mismatch in this statement. FCI learns equivalence classes of MAGs, while $\mathcal{G}$ is a DAG and $\mathcal{G}^{\setminus \mathbf{U}}$ is an ADMG.

- Algorithm 1: In the definition of $\mathbf{A}$, the condition $X \preceq {X_i, X_j}$ is meant to be with respect to the set of already identified ancestral relationships as encoded in $\mathbf{M}$, right? Please specify this explicitly in the final version in order to avoid confusion.

- Algorithm 1 continued: The symbol $\mathcal{I}$ that surrounds the independence statements is not explained.

- "By contrast, for an empty $\mathbf{Z}$, CBL-ORACLE will not provide any information": It would still find $X_1 \sim X_2$, wouldn't it? I suppose you are saying that it does not find any ancestorships.

- $X_{\preceq i}$ and Theorem 2: I suppose you want $X_i \in \mathbf{X}_{\preceq i}$, right? Please indicate this explicitly in order to avoid confusion.

- Theorem 2 continued: It would be helpful to show one graph in which these assumptions are fulfilled and one graph in which they are not fulfilled, perhaps accompanied by a short exemplification of Theorem 2 with regard to these two graphs. This could be placed in the supplementary material.

- "Observe that inferences about ancestral relationships are fully ordered with respect to their information content": Is $i \preceq j$ or $i \sim j$ more informative? I would think neither of them. Does this pose a problem for your concept of "Dominance"?

- "This is well-understand in, for example ... even if the corresponding conditional independence does not hold": Please add references.

- "If either event occurs fewer than $\gamma B$ times (with the convention that $\gamma = 1 / 2 $) we conclude that $X \sim Y$": I realize I am being a bit pedantic here, but is this really what Algorithm 3 is doing? The sentence reads like the two events are checked independent of each other for each of the $B$ repetitions, while Algorithm 3 for each repetition checks whether neither of them occurs.

- Theorem 4: Is $D(\theta, \tau, B, r)$ known? It seems to required in Algorithm 2.

- "Constraint-based methods show less power ...": This seems to be quite a generalization given that the experiment encompasses only one constraing-based method.

- Section 6.3: Should it perhaps be $E[\epsilon_X | \mathbf{Z}]$ and $E[\epsilon_Y | \mathbf{Z}, X]$ instead of $E[\epsilon_X | \mathbf{Z}, X]$ and $E[\epsilon_Y | \mathbf{Z}]$?

**Q7 Justification For Your Score:**

The strenghts clearly outweigh the relatively minor weaknesses and I do not have conceptual concerns. I was even considering scoring the paper as "strong accept".

**Q9 Complying With Reviewing Instructions:**

1: Yes.

---

### Official Review · Reviewer_g8Wy · 2022-04-10

**Q2(1) Originality/Novelty:** 2
**Q2(2) Significance/Impact:** 3
**Q2(3) Correctness/Technical Quality:** 3
**Q2(6) Clarity Of Writing:** 3
**Q6 Overall Score:** 7
**Q8 Confidence In Your Score:** 3

**Q1 Summary And Contributions:**

The paper aims to recover a directed acyclic subgraph known to be causally descended from a confounder blanket without parametric restrictions or extreme sparsity constraints. The three rules of the paper are the same as the paper of Entner et al.  2013. Therefore,  it seems to be an extension of the three rules by using the known partial causal information $\mathbf{Z}$ for recovering the directed acyclic subgraph in $\mathbf{X}$.

**Q2 Assessment Of The Paper:**

More detailed information regarding each of these aspects is given below:

**Q2(4) Quality Of Experiments (Optional):**

3: Good: The experimental evaluation is adequate, and the results convincingly support the main claims.

**Q2(5) Reproducibility:**

3: Good: Key resources (e.g., proofs, code, data) are available and key details (e.g., proofs, experimental setup) are sufficiently well-described for competent researchers to confidently reproduce the main results.

**Q3 Main Strengths:**


1. A sound and complete set of rules for inferring causal order in sub-DAGs with a confounder blanket.
2. An algorithm, CBL-sample is developed for efficient and flexible on linear and nonlinear systems.
3. Experiments on synthetic and real-world data confirm the performance of the developed CBL-sample with high power and bounded error.

**Q4 Main Weakness:**

1. The confounder blanket seems to be a stronger assumption than the pretreatment assumption that is always made in the literature of causal inference as Entner et al. 2013. The paper can be regarded as an extension of Entner et al. 2013. It would be better to discuss the similarities and differences between the two works.
2.  The experimental results in Figure 2 are the same as the results in Entner et al. 2013. There are no new conclusions.
3. M-structure is an important problem when there are latent variables in causal inference, it is not clear to see How can the developed algorithm deal with M-bias.


**Q5 Detailed Comments To The Authors:**

The presented statistic inference is interesting and useful. There are two concerns as below:
*"The first assumption can be relaxed, since we concerned primarily with subgraph $\mathcal{G}_X$, …" If there is a direct cycle in subgraph $\mathcal{G}_X$, will Markov properties still hold? I found that the work assumes that a DAG $\mathcal{G}$ contains $\mathbf{Z}\cup \mathbf{X}$. If $\mathcal{G}_X$ contains a cycle, then the assumption of a DAG $\mathcal{G}$ will be violated such that it seems not a problem of  "further technicalities".
*In Definition 3, “do not attempt to test for conditional independencies using arbitrary subsets of $\mathbf{Z}$. if some $\mathbf{A}\subseteq \mathbf{Z}\cup \mathbf{X}$ is a valid adjustment set for any ordered pair $X_i\prec X_j$, then $\mathbf{A} \cup \mathbf{Z}$ is also valid.”, It seems to be  $\mathbf{A}\subseteq \mathbf{X}$ according to the previous statement.

**Q7 Justification For Your Score:**

The main contribution of the work is from Entner et al. 2013,  and the soundness and completeness are deduced from the conclusion of Entner et al. 2013. Incredibly, the experimental design is also very similar to Entner et al. 2013's. Hence, I can not fully support this work and further clarification is required before I can raise my score.

**Q9 Complying With Reviewing Instructions:**

1: Yes.

---

### Decision · Program_Chairs · 2022-05-15

**Decision:**

Accept (Poster)

**Comment:**

Meta Review: **Summary:** There was remarkable agreement among reviewers, with all indicating the paper introduced a novel but intuitive approach, tackling a relevant problem with solid theoretical support, and was, above all, exceptionally well written.
I agree with reviewers it is an interesting and well written article that clearly deserves to be in the conference. Decided on'accept as poster' but could be an oral as well.

**Caveat:** the ‘sound and complete’ algorithm claim in the abstract could be considered misleading, as it turns out (Thm.3) this is only ‘relative to a lazy oracle’. The paper tries to get away with this by calling it ‘a new notion that places reasonable limits on completeness’ (p4,above Def.4), but this is not what is typically understood by an unqualified completeness claim.

**Quality:** High quality paper with a variety of solid theoretical results, and good experimental support.

**Clarity:** Very readable with good explanations.

**Originality:** Original approach that extends (Entner 2013).

**Significance:** Perhaps a tad niche (the approach is tailored to very dense, high-dimensional networks), but with some solid theoretical results that will be of interest to the (causal) community.